# A New SNP in Rice Gene Encoding Pyruvate Phosphate Dikinase (PPDK) Associated with Floury Endosperm

**DOI:** 10.3390/genes11040465

**Published:** 2020-04-24

**Authors:** Heng Wang, Tae-Ho Ham, Da-Eun Im, San Mar Lar, Seong-Gyu Jang, Joohyun Lee, Youngjun Mo, Ji-Ung Jeung, Sun Tae Kim, Soon-Wook Kwon

**Affiliations:** 1Department of Plant Bioscience, College of Natural Resources and Life Science, Pusan National University, Miryang 50463, Korea; wang_heng126@126.com (H.W.); ekdms0309@gmail.com (D.-E.I.); sanmarlar2010@gmail.com (S.M.L.); sgjang0136@gmail.com (S.-G.J.); stkim5505@gmail.com (S.T.K.); 2State Key Laboratory of Microbial Metabolism, School of Life Sciences and Biotechnology, Shanghai Jiao Tong University, Shanghai 200240, China; 3Department of Applied Bioscience, Konkuk University, Seoul 05029, Korea; lion78@daum.net (T.-H.H.); joohyun00@gmail.com (J.L.); 4National Institute of Crop Science, Rural Development Administration, Jeonju 54874, Korea; moyj82@korea.kr (Y.M.); jrnj@korea.kr (J.-U.J.)

**Keywords:** *Oryza sativa* L., PPDK, *flo4-5*, floury endosperm

## Abstract

Rice varieties with suitable flour-making qualities are required to promote the rice processed-food industry and to boost rice consumption. A rice mutation, Namil(SA)-flo1, produces grains with floury endosperm. Overall, grains with low grain hardness, low starch damage, and fine particle size are more suitable for use in flour processing grains with waxy, dull endosperm with normal grain hardness and a high amylose content. In this study, fine mapping found a C to T single nucleotide polymorphism (SNP) in exon 2 of the gene encoding cytosolic pyruvate phosphate dikinase (cy*OsPPDK*). The SNP resulted in a change of serine to phenylalanine acid at amino acid position 101. The gene was named *FLOURY ENDOSPERM 4-5* (*FLO4-5*). Co-segregation analysis with the developed cleaved amplified polymorphic sequence (CAPS) markers revealed co-segregation between the floury phenotype and the *flo4-5*. This CAPS marker could be applied directly for marker-assisted selection. Real-time RT-PCR experiments revealed that PPDK was expressed at considerably higher levels in the *flo4-5* mutant than in the wild type during the grain filling stage. Plastid ADP-glucose pyrophosphorylase small subunit *(AGPS2a* and *AGPS2b)* and soluble starch synthase (*SSIIb* and *SSIIc)* also exhibited enhanced expression in the *flo4-5* mutant.

## 1. Introduction

Rice (*Oryza sativa* L.) is a staple food for more than half the world’s population; improving yield and grain quality is therefore of great importance. Grain yield can be increased considerably through the use of hybrid rice strains, but the grain quality of these varieties does not fully satisfy customer demands [1]. As such, grain quality improvement is the current focus of many rice geneticists worldwide. Furthermore, cultivars with a range of different grain qualities are desirable for specific production or medicinal purposes. 

Starch, which is the major storage carbohydrate in rice grains, constitutes approximately 90% of a rice grain, of which approximately 18% is amylose and 82% is amylopectin. The ratio of amylose to amylopectin plays an important role in rice grain structure, appearance, and eating quality. Amylose content is the most important determining factor for almost all of the physicochemical properties of rice starch, including its gelling, pasting, firmness, turbidity, freeze–thaw stability, syneresis, and retrogradation properties [2,3]. Amylopectin content governs the formation of crystalline granules and paste thickening [4]. Much research in recent decades has focused on understanding the genetics and biochemistry of starch biosynthesis, which is a complex process through which glucose generated by photosynthesis is converted to starch via several intermediaries. AGPase catalyzes the first committed step of starch biosynthesis and, as such, plays an important role in regulating starch synthesis in rice endosperms. Rice grains from plants carrying mutations in the AGPase gene exhibit shrunken endosperms and reduced starch content as a result of significant reductions in starch synthesis [5,6]. By contrast, the overexpression of AGPase genes increases seed weight and starch content [7]. Both amylose and amylopectin utilize ADPG as the activated glucosyl donor, and subsequent synthesis steps use different enzymes for amylose and amylopectin production. Amylose is synthesized by *GBSSI* (granule-bound starch synthase), which is encoded by *Wx* (waxy). Loss-of-function mutations at the *GBSSI*/*Wx* locus result in the elimination of amylose synthesis, leading to a lack of long-chain amylopectin, as well as a complete lack of amylose [8,9].

Starch synthesis is also transcriptionally regulated by transcription factors that presumably function within biosynthetic networks [1]. These include alkaline leucine zipper transcription factor OsbZIP58, which regulates *OsAGPL3*, *Wx*, *OsSSIIa*, *SBE1*, *OsBEIIb*, and *ISA2* transcription through binding to their promoters [10]. A lack of *ISA* in the rice sugary mutant impacts the expression of several other genes related to starch synthesis [11]. The high-resistant starch (RS) mutant, which is defective in SSIIIa, affects the activity of PPDK (pyruvate phosphate dikinase) and AGPase and leads to increases in lipid and amylose synthesis [12]. The white-core endosperm mutant, which is defective in *OsPK2*, affects the expression of genes involved in glycolysis/gluconeogenesis, pyruvate/phosphoenolpyruvate metabolism, fatty acid metabolism, and starch synthesis [13].

Among endosperm mutants, only those with a floury and white core contain round and loosely packed starch granules that grind easily [14]. To date, seven floury endosperm genes (*FLO1–7*) have been identified in rice [15,16,17,18,19,20,21], six of which have been cloned and their functions characterized. Three of these genes are not directly involved in starch synthesis and are instead involved in protein–protein interactions that regulate starch-synthesis-related genes. *Flo2* encodes a tetratricopeptide repeat motif that affects *Wx* expression, Flo6 encodes a C-terminal carbohydrate-binding module 48 domain that regulates isoamylase1, and Flo7 harbors an N-terminal transit peptide that does not directly interact with starch synthesis genes [19,20,21]. The three remaining floury endosperm genes participate in starch synthesis pathways; Flo4 encodes a PPDK, Flo5 encodes a rice-soluble starch synthase gene (*OsSSIIIa*), and *osagpl2–3* (also named *Flo6*) encodes AGPase [15,18,22]. In rice starch metabolism pathways, amylopectin-synthesizing enzymes, including starch synthases (*SSIIIa*, *SSIVb*, and *SSIIa*), starch-branching enzymes (*SBEI* and *SBEIIb*), and *PUL*, physically interact with each other to form multienzyme complexes [23,24]. The multienzyme complexes also contain other enzymes, such as PPDK and AGPase, that control the partitioning of ADP-Glc into starch and lipid [12,24,25]. This suggests that the proteins encoded by the three floury endosperm starch synthesis genes could interact with one another. The PPDK encoded by *Flo4* could promote AGPase activity in the complex by directly supplying PPi (pyrophosphate) for the conversion of ADP-Glc to Glc-1-P, which would enhance lipid biosynthesis. *SSIIIa* could inhibit the activity of PPDK/AGPase [25], and this interaction could be disrupted in the *SSIIIa* mutant (G to A in an exon), releasing PPDK and AGPase and increasing lipid synthesis. Simultaneously, loss of *SSIIIa* activity could divert the carbon flow needed for amylose synthesis by *GBSSI* (encoded by *Wx*), and as a result, the balance of lipid and amylose, which are components of RS, could be disrupted [12,25]. For AGPase, an SNP in the open reading frame (ORF) region of *osagpl2–3* may impair the activity of the functional domain that interacts with ADP-Glc. The effect of this mutation is particularly apparent at the endosperm development stage and results in the mutant floury endosperm phenotype [22]. Nevertheless, direct evidence of PPDK’s function in the multienzyme complex remains unknown.

Rice has three PPDK-encoding genes organized at two loci: PPDKA, which encodes a cytosolic isoform (*OsPPDKA*), and PPDKB, which has two promoter sites and produces cytosolic and chloroplastic enzymes (cyPPDKB and chPPDKB, respectively) [8,15,26]. Of these, cyPPDKB is critical during the filling stage of rice grain development. During this stage, cyPPDKB is produced in abundance and acts to provide carbon skeletons for amino acid and lipid synthesis through the reversible interconversion of pyruvate and Pi to phosphoenolpyruvate and PPi. As seed development progresses from the early stage to the final storage product accumulation stage, the level and activity of cyPPDKB decrease rapidly in response to the combined posttranslational mechanisms of threonyl phosphorylation and protein degradation [27]. Cytosolic PPDKB is also involved in the glycolytic and gluconeogenic pathways and is impacted by anoxia [28,29,30,31]. To determine PPDK function in developing seeds, experiments to transgenically eliminate the PPDK gene from rice endosperms, to release PPDK from the phosphate-sufficient condition, or to upregulate PPDK expression during the middle and late rice grain-filling stages would be logical. However, the first report of a T-DNA insertional knockout mutant of the rice PPDK gene (*flo4-1*, *flo4-2* and *flo4-3*) indicated that rice with inactivated PPDK produce unexpected opaque seeds with a high lipid content [15].

The Namil(SA)-flo1 rice mutant was developed via sodium azide mutagenesis of *Oryza sativa* ssp. japonica cv. Namil [32]. By screening the mutant stock of Namil, two allelic mutants exhibiting floury endosperms were isolated and named “Namil(SA)-flo1” and “Namil(SA)-flo2” (Suweon 542). Suweon 542 exhibited a milky-white opaque endosperm, except for a thin peripheral area. Physicochemical analysis of the Suweon 542 endosperm revealed a loosely packed structure of irregular and globular-shaped starch granules, a low protein content and grain weight, and high amylose content compared with its wild-type progenitor, Namil. During dry milling, Suweon 542 had significantly lower grain hardness, finer particle size, more loosely packed starch granules, and lower starch damage than Namil and other rice cultivars [33]. Genetic analysis of the floury endosperm characteristics of Suweon 542 revealed that the location of the target gene was in the 19.33–19.73 Mbp region on chromosome 5 between markers RM18624 and RM18639, and map-based cloning revealed a G→A SNP in exon 8 of cy*OsPPDK* (*flo4-4*), which was responsible for the floury endosperm of Suweon 542 [33,34]. Moreover, the floury endosperm of Namil(SA)-flo1 controlled by one recessive gene and the locus of Namil(SA)-flo1 was localized to the 17.7–20.7 Mbp region of chromosome 5 [35]. However, the grain characteristics and molecular mechanisms regulating the floury endosperm in Namil(SA)-flo1 remain unknown. Fine mapping and molecular cloning of floury endosperm-related genes are needed to verify and elucidate the genetic framework of floury endosperm development in rice.

In this study, the agronomic traits and grain physicochemical properties, including suitability for dry milling of Namil(SA)-flo1, were investigated. Map-based cloning revealed that a C→T SNP in exon 2 of cy*OsPPDK* results in a missense mutation from Ser to Phe at amino acid position 101. Co-segregation analysis and qRT-polymerase chain reaction (PCR) confirmed that *flo4-5* was responsible for the development of the floury endosperm during the grain filling stage and indicated the involvement of cy*OsPPDK* in grain quality and seed number control.

## 2. Materials and Methods

### 2.1. Plant Materials and DNA Extraction

The Namil(SA)-flo1 floury endosperm rice mutant was produced via sodium azide mutagenesis of Namil (*O. sativa* L. ssp. japonica), an early maturing, high-yield, non-waxy Korean elite rice cultivar [12]. To evaluate the major agronomic traits and the grain/flour physicochemical properties, Namil(SA)-flo1 was cultivated with Namil, Seolgaeng (non-waxy opaque endosperm japonica cultivar), and Hwaseong (non-waxy japonica cultivar) in the experimental plot of the National Institute of Crop Science (NICS), Rural Development Administration (RDA), Suwon, Korea.

### 2.2. Evaluation of Agronomic Traits and Grain Physicochemical Properties

Replicated yield trials were conducted to evaluate the major agronomic traits, as well as the grain/flour physicochemical properties, in the field at the NICS, RDA, Suwon, Korea [35]. The seeds of each rice line were sown on April 25, and were then transplanted on May 25 under a randomized complete block design (RCBD) with three replication plots. Each plot, consisting of eight rows with 30 hills per row and three plants per hill, was planted with 30 × 15 cm spacing. The amount of fertilizer application was 90–45–57 Kg/ha for N–P_2_O_5_–K_2_O, and the 10 hills in the middle rows were used to determine days-to-heading (HD), culm length (CL), panicle length (PL), tiller number (TN), spikelet number per panicle (SN), and ripened grains percentage (RGP). The 1000-grain weight (TGW) was measured in grams as the average weight of 1000 fully filled brown rice grains from each plot.

The grain hardness of the brown rice was assessed by determining the pressure at the grain breakage point using a 5 mm probe attachment of a TA.XT Plus instrument (Stable Micro Systems, Godalming, Surrey, UK), using parameters of 0.4 mm/s and 40.0 g for test speed and trigger force, respectively.

The damaged starch content was evaluated using a starch damage assay kit (Megazyme International Ireland, Wicklow, Ireland) following the manufacturer’s instructions. The lightness of rice flour was measured with a JS-555 instrument (Color Techno System, Tokyo, Japan). The moisture, protein, lipid, and ash contents of the rice flour were determined using methods 44-15A, 46-30, and 08-01 of the American Association of Cereal Chemists (AACC) 2000. The amylose content of the rice flour was estimated as described previously [36].

Grain samples for scanning electron microscopy (SEM) analyses were prepared as described previously [15]. Cleaved endosperm surfaces were observed under an S-550 scanning electron microscope (Hitachi Hi-Tech, Tokyo, Japan) at an accelerating voltage of 20 kV.

### 2.3. Fine Mapping of FLO4-5

In a previous study, the F2 population of Namil(SA)-flo1/Milyang 23 (non-waxy Tongil cultivar) was used for genetic analysis [33]. To investigate the candidate genomic region for *FLO4-5*, F_2_ lines heterozygous for the genomic region between RM18624 and RM18639 were grown to the F_3_ generation. The F_3_ plants were cultivated to generate F_3:4_ seeds at the Pusan National University (PNU) experimental farm. Total genomic DNA was extracted from fresh leaves using a NucleoSpin® Plant II kit (MACHEREY-NAGEL GmbH & Co.KG, North Rhine-Westphalia, Germany), according to the manufacturer’s instructions.

Parental lines and 96 randomly selected F_3:4_ recombinants were dehulled for visual inspection of the endosperm, and then genotyped with six cleaved amplified polymorphic sequence (CAPS) markers developed based on the whole-genome resequencing data [34] of Namil(SA)-flo1 and Milyang 23. The whole genomes of Namil(SA)-flo1 and Milyang 23 were re-sequenced with a 75-fold average coverage using the Illumina HiSeq 2500 Sequencing Systems Platform (Illumina Inc., San Diego, CA, USA). Raw sequence reads were aligned against the rice reference genome (IRGSP 1.0) [37], and the predetermined CAPS marker orders were judged by e-landing of each marker on the reference rice genome [38]. ORFs and their functional products were annotated according to the MSU Rice Genome Annotation Project Database [39] based on the defined physical locations. PCRs were performed in a total volume of 20 μL containing 10 ng of DNA template, 10 pmol of each primer, 1× PCR buffer (50 mM KCl, 10 mM Tris-HCl (pH 9.0), 0.1% Triton X-100, and 1.5 mM MgCl_2_), 0.2 mM dNTPs, and 1 unit of Taq DNA polymerase (Nurotics, Daejeon, Korea). PCRs were performed with an MJ Research PTC-100 thermocycler (Waltham, MA, USA) using the following conditions: initial denaturation at 94 °C for 5 min, followed by 36 cycles of denaturation at 94 °C for 30 s, annealing at 58 °C for 30 s, and extension at 72 °C for 1 min, with a final extension at 72 °C for 10 min. The PCR products were digested using restriction enzymes (New England Biolabs, Ipswich, MA, USA), according to the manufacturer’s instructions. The digestion of PCR products encompassing CAPS markers was detected using the WatCut program [40]. Digestion products were separated on 3% polyacrylamide gels using 6 M urea and 1X TAE at 80 volts and visualized using the Molecular Imager® Gel Doc™ XR System (Bio-Rad Laboratories, Inc., Hercules, CA, USA).

### 2.4. Cloning of FLO4-5 and Identification of the Mutation Site

The coding sequence of cy*OsPPDK* in Namil(SA)-flo1 and Namil was compared using the CLC Sequence Viewer 7.0 (QIAGEN, Hilden, Germany). To verify the mutation site, a CAPS marker containing two *MboII* restriction sites in the floury endosperm mutant Namil(SA)-flo1 and one *MboII* restriction site in Namil was developed using the WatCut program. A 177 bp fragment containing the mutation site was PCR amplified from Namil(SA)-flo1 and Namil using the primers flo4-5_F: CTCCAGTGGGTGGAGGAGTA and flo4-5_R: GATCGATCAGCAACGGAGAT. The PCR products were digested with *MboII* (New England Biolabs) in a volume of 15 μL containing 5 μL of PCR product, 1.5 μL of 10× NEBuffer, 0.5 μL of *MboII*, and 8 μL of ultrapure water, and then incubated at 37 °C for 2 h. The digestion products were separated using the Fragment Analyzer™ (Agilent, Santa Clara, CA, USA). The CAPS marker was also used for co-segregation analysis of F_3:4_ families and 44 Korean rice cultivars with the endosperm phenotype. The complete genomic DNA of the cy*OsPPDK* gene was cloned in three overlapping segments using primers designed on the basis of the cy*OsPPDK* gene sequence of Nipponbare (Appendix A).

### 2.5. Predicting the Functional Effect of Amino Acid Substitutions and Real-Time qRT-PCR Analysis

The PROVEAN (Protein Variation Effect Analyzer, v1.0) tool [41] was used to predict amino acid changes that affected protein function. Total RNA was isolated from the rice grains of Namil and Namil(SA)-flo1 at 12 days after flowering (DAF) using an RNeasy Plant Mini Kit (QIAGEN, Hilden, Germany), according to the manufacturers’ instructions. Genomic DNA was removed with DNase I (QIAGEN), and reverse transcription was performed using an RNA to cDNA EcoDry Premix Kit (Clontech, Mountain View, CA, USA). A QuantiNova SYBR Green RT-PCR kit (QIAGEN) and the Rotor-Gene Q instrument (QIAGEN) were used for qRT-PCR with the following conditions: 95 °C for 10 min followed by 40 cycles of 95 °C for 10 s and 60 °C for 20 s. Fold-change was calculated relative to Namil. Rice Actin 1 (OsACT1; Os03g0718150) was used as an internal control. The primer sequences used for qRT-PCR are listed in Appendix A.

## 3. Results

### 3.1. Agronomic Traits and Seed Characteristics of Namil(SA)-flo1

The major agronomic and yield-related traits of the Namil(SA)-flo1 mutant and the corresponding Namil wild type were analyzed in plants grown under paddy field conditions. Compared with the wild type, heading was delayed by 3 days and CL was extended by 4 cm in the Namil(SA)-flo1 mutant. Panicle size, tiller number, and spikelets per panicle did not differ significantly between the wild type and Namil(SA)-flo1 mutant, but RGP was lower in Namil(SA)-flo1 than in the wild type (Table 1).

Cross-sectional observation of dehulled kernels revealed that most of the Namil(SA)-flo1 endosperm was white–opaque, except for a thin peripheral area (Figure 1a). SEM showed that the Namil(SA)-flo1 endosperm contained numerous small starch grains of irregular and rounded shapes that were more loosely packed than in the wild type. The wild-type endosperm exhibited densely packed starch granules of polyhedral angular shapes (Figure 1b). In contrast with the appearance of the grain and the endosperm in Namil, Namil(SA)-flo1 exhibited a milky-white opaque endosperm. No differences in grain width, length, or thickness were observed between the mutant and the wild type (Figure 1c).

The flour physicochemical properties were also examined. The results show that the flour derived from the mutant grains was significantly lighter than that from the wild-type grains. The ash content was lower in Namil(SA)-flo1 than in the wild type. The amylose content was comparable between the wild type and Namil(SA)-flo1, but the lipid content was higher and the protein content was significantly lower in Namil(SA)-flo1 than in the wild type (Table 2). These findings are consistent with the results of previous reports showing that increasing the lipid content could cause a floury endosperm [12].

### 3.2. Dry Milling Properties of Namil(SA)-flo1

The suitability of Namil(SA)-flo1 as a raw material for high-quality dry milled rice flour was assessed. The grain hardness of Namil(SA)-flo1 was 0.45 times that of the wild type, and was softer than that of Seolgaeng, a Korean rice cultivar with an opaque endosperm. The mean particle size of Namil(SA)-flo1 was 86.1 μm, smaller than that of the wild type (109.1 μm), Seolgaeng (97.6 μm), and Hwaseong (112.2 μm). The damaged starch content of dry milled flour in Namil(SA)-flo1 (4.9%) was significantly lower than that in Namil (9.2%), Seolgaeng (7.1%), and Hwaseong (10.3%) (Table 2). Overall, the low grain hardness, low starch damage, and fine particle size indicate that Namil(SA)-flo1 is suitable for dry milling to produce fine flour.

### 3.3. Fine Mapping of the Floury Endosperm Locus

The Namil(SA)-flo1 locus was previously refined to a region on chromosome 5 and, alongside *FLO4-4* (identified from Namil(SA)-flo2, also known as Suweon 542), was linked to chromosome 5 markers RM18624 and RM18639 [33,35]. To map the Namil(SA)-flo1 target gene more precisely, the whole genomes of Namil(SA)-flo1 and Milyang 23 were re-sequenced and aligned against one another to select homologous SNPs for developing CAPS markers. Six CAPS markers were developed (Appendix A) and used to genotype 60 floury and 36 normal individuals derived from F_3:4_ recombinant families. Multiple comparisons were conducted between the genotypes of these recombinants and the phenotypes of their offspring. Finally, the target region was mapped to BAC clones OJ1174_H11 and BAC OSJNBb0006J12 in a 33 kb region flanked by the markers CAPS6 and RM18639. Based on the Rice Annotation Project Database [42], four predicted genes were found in the critical 33 kb region (Figure 2a). Of these, *Os05g0404500* was annotated as encoding a hypothetical protein, Os05g0404700 was a functional gene similar to the methyl-binding protein gene MBD1, Os05g0404901 encoded a conserved hypothetical protein, and Os05g0405000 was annotated as a PPDK gene. Sequencing of all four candidates in Namil(SA)-flo1 and the Namil wild type revealed a C to T SNP in exon 2 of PPDK (Figure 2b and Appendix A). In rice, PPDK, which has two promoter sites, encodes both a cytosolic- and a chloroplast-targeted PPDK protein (cy*OsPPDK* and ch*OsPPDK*, respectively). To analyze the transcript type of PPDK in developing grains of rice, the PPDK of Namil(SA)-flo1 and Namil was sequenced using the primer combinations PF2/PR3 (cy*OsPPDK*) and PF3/PR3 (ch*OsPPDK*) [15] and three additional primer pairs (Appendix A). The full-length sequence of cy*OsPPDK* was identified as 7556 bp with a cDNA sequence of 2649 bp containing 19 exons and encoding an 882 amino acid protein. The SNP in exon 2 was a missense mutation that resulted in a serine to phenylalanine change at amino acid position 101 in cy*OsPPDK* (Figure 2b). Comparison of the cy*OsPPDK* sequences of Namil(SA)-flo1 and Namil with that of Nipponbare revealed nine SNPs and six InDels (insertion/deletion mutations) in the intron region and two SNPs in the coding region that were synonymous mutations (Appendix A). Sequence analysis of Suweon 542 (*flo4-4*) did not reveal the same mutation site (the Suweon 542 phenotype was caused by a G→A SNP in exon 8 of cy*OsPPDK*) [34]. The novel recessive floury gene was named *floury4-5* (*flo4-5*). The full-length coding sequences of *FLO4-5* from Namil(SA)-flo1 and Namil were deposited in GenBank under accession numbers MG267057 and MG267056, respectively.

### 3.4. Co-Segregation and Expression Analyses

The mutation site in *flo4-5* was confirmed using a CAPS marker with a *MboII* restriction site. *MboII* digested the mutant Namil(SA)-flo1 allele at two restriction sites (Figure 3a,b, lane 3) and the wild-type Namil allele at a single site (Figure 3a,b, lane 4). For screening F_3:4_ family populations using the CAPS marker, 23 plants with the Namil(SA)-flo1 phenotype and 22 with the Namil phenotype were selected. As shown in Figure 3a, sequences from all plants with the Namil(SA)-flo1 phenotype were digested at two restriction sites with *MboII*, whereas those with the Namil phenotype were digested at a single site. Fifteen F_3:4_ plants were heterozygous and exhibited the Namil phenotype, suggesting that the floury endosperm phenotype was controlled by a recessive gene. The genotypes of the currently cultivated Korean japonica rice were evaluated with this marker. Of the 44 tested varieties, none of them were the mutant genotype, whereas two varieties were the Namil genotype and the others were the Nipponbare genotype (Figure 3b). Overall, the co-segregation analysis indicates that *FLO4-5* is responsible for the floury endosperm in rice.

The PROVEAN tool was used to predict the effect of a single nucleotide mutation (C to T) on the *FLO4-5* gene, resulting in a score of –5.802 and a rating of “deleterious”. Next, qRT-PCR was used to investigate the transcript type of *FLO4-5* in rice during the grain filling stage. The result that relative expression of *FLO4-5* was much higher in Namil(SA)-flo1 than in Namil (Figure 4) was unexpected, because the mutation was in exon 2, not in the promotor region. Even though the higher expression level in the mutant was not clearly explained based on our experimental design, it is highly possible that the increased level of *FLO4-5* was caused by the mutation.

### 3.5. Mutation of FLO4-5 Changed the Expression Levels of Major Starch Synthesis Enzymes

Because the starch content and the starch granules were altered in the *flo4-5* mutant, the expression levels of the starch synthesis-related genes in the developing grains were then analyzed using qRT-PCR. Compared with the wild-type Namil, the transcription of several genes decreased in the mutant, including *GBSSI*, *GBSSII*, *AGPL1*, *AGPL2*, *AGPL3*, *AGPL4*, *AGPLS1*, *SSI, SSIIa, SSIIIa, SSIIIb, SSIVa, SSIVb, BEI, BEIIa, BEIIb,* and *PUL*. The expression of *AGPS2a*, *AGPS2b*, *SSIIb*, and *SSIIc* increased in the mutant compared to the wild type, with substantial increases seen for *AGPS2a* and *AGPS2b* (Figure 5a). The expression analysis of Suweon 542 (*flo4-4*) revealed similar expression patterns (Figure 5b).

## 4. Discussion

Cereal crops accumulate high levels of starch in the seed endosperm as an energy reserve. Endosperm-defective mutants provide valuable genetic material for elucidating the gene networks that regulate starch synthesis and amyloplast development during grain filling. In this study, we isolated and characterized the Namsil(SA)-flo1 mutant, which exhibited an almost entirely milky-white opaque kernel, except for a thin peripheral area of the grain (Figure 1a). Compared to the wild type, the Namsil(SA)-flo1 mutant exhibited slightly higher amylose levels, lower protein levels, reduced kernel weight, and increased total lipid content (Table 2). No difference in seed size was observed (Figure 1c). Electron microscope visualization of the transverse sections of the Namsil(SA)-flo1 and wild-type mature endosperms showed that the Namsil(SA)-flo1 mutant was loosely packed with irregular and round-shaped compound starch granules, while the wild type contained densely packed polyhedral starch granules (Figure 1b). However, the physicochemical characteristics of Namsil(SA)-flo1 were different from those reported for *flo4-1*, *flo4-2,* and *flo4-3*, which were generated via T-DNA insertion into the cy*OsPPDK* gene [15,33,34]. We speculate that the cy*OsPPDK* gene may have diverse roles in the rice seed filling stage. Fine mapping, sequencing, and co-segregation analysis revealed that a C to T SNP in exon 2 of cy*OsPPDK* (the novel recessive floury gene was named *flo4-5*) was responsible for the Namsil(SA)-flo1 mutant phenotype (Figure 2 and Figure 3). Based on our limited expression analysis method, this is not clearly explained; however, we hypothesize that the transcript from the mutated sequence compared to that of the normal sequence may not produce normal PPDK to increase the transcription level in the mutant, in order to compensate the normal transcript concentration in the cell. This hypothesis will be proven in a further study. Along with an increase in the transcripts of PPDK/AGPase, the expression of changes for starch synthesis genes (Figure 5) suggests that PPDK has a role in compound starch granule formation and starch synthesis during rice endosperm development.

PPDK catalyzes the freely reversible conversion of pyruvate, ATP, and Pi into phosphoenolpyruvate, AMP, and PPi, and is involved in diverse functions in various plant tissues. PPDK is ubiquitous in all rice tissues, but is generally of low abundance [43]. In developing rice kernels, PPDK is abundantly expressed during the syncytial/cellularization stage, with markedly reduced expression seen during the middle and late rice grain filling stages [15,27]. The presence of PPDK in high abundance in developing seeds suggests the importance of the enzyme; however, little is known regarding the molecular properties of rice PPDK. 

The interconversion of phosphoenolpyruvate and pyruvate by PPDK leads to difficulties in evaluating PPDK function in seed development [43]. Previous studies have suggested several hypotheses for endosperm PPDK function, including (1) provision of pyruvate for lipid synthesis, (2) gluconeogenesis and provision of hexose for starch biosynthesis, (3) control of metabolic fluxes through its contribution to PPi homoeostasis, and (4) regulating glycolytic flux and energy charge [24,27,28,30,44,45]. In *Zea mays,* the PPi generated by PPDK can be channeled directly to AGPase within the protein complex, driving the plastidial AGPase reaction in the direction of ADPGlc breakdown to Glc-1-P, which can in turn support amino acid and lipid biosynthesis [24]. However, a loss-of-function mutation in the rice cy*OsPPDK* gene resulted in a lower amylose content, reduced kernel weight, increased total lipid content, and no change in storage protein or amino acid contents [15]. This suggests that PPDK may have several diverse roles in seed development, depending, in part, on the direction of the cycle [43]. 

In amylopectin biosynthesis, *SS* connects glucose in an α-1,4 glycosidic bond formation, and then *BE* (branching enzyme) connects a short chain to the C-6 hydroxyl group via hydrolysis of internal α-1,4 glycosidic bonds and formation of α-1,6 branches. *DBE* (starch-debranching enzyme: Isoamylase (*ISA*) and pullulanase (*PUL*)) cleaves aberrant branches through catalytic hydrolysis of α-1,6 glycosidic bonds. A lack of *SS* during rice endosperm amylopectin formation results in a decrease in polymerized chain formation and an increase in amylose characteristics [12,46,47,48,49]. Mutant *ae*, which is defective in the gene encoding *BE*, exhibits high proportional levels of amylose as a result of reductions in amylopectin production [50,51]. A loss-of-function mutation of the gene encoding *DBE* results in a sugary phenotype in which endosperm starch is completely replaced by phytoglycogen [52,53]. Mutant *sug-h*, which is defective in *OsISA1* and *OsBEIIa*, exhibits a severely disrupted amylopectin structure [54].

A recent study revealed that rice amylopectin-synthesizing enzymes such as *SSI, SSIIIa, BEI, BEIIa, BEIIb, ISA1,* and *PUL* are physically associated with each other and form active protein complexes with various partnership patterns [12,23]. Further research in maize and rice indicates that PPDK and plastid AGPase may also participate in these enzyme complexes [12,24]. In these complexes, PPDK catalyzes the freely reversible conversion of pyruvate, ATP, and Pi into phosphoenolpyruvate, AMP, and PPi, but in photosynthesis, PPDK generates phosphoenolpyruvate by the hydrolysis of PPi used for CO_2_ fixation in C4 plants [24]. Plastid AGPase also catalyzes a reversible reaction, but the direction of the reaction only depends on the relative concentrations of PPi and ATP [30]. Therefore, high PPi concentrations should orient the reaction to the direction of ADP-Glc degradation. However, the PPi levels in amyloplasts are very low as a result of high pyrophosphatase (PPase) activity, and the reaction therefore proceeds toward ADP-Glc synthesis. Zhou et al. [12] hypothesized that the starch biosynthetic enzymes in the complex might exert a constraining effect on PPDK and AGPase to control the partitioning of ADP-Glc into lipid and starch. 

Based on previous research and our findings, we hypothesize that a single amino acid change (S to F) in the PPDK protein in the flo4-5 mutant may disrupt the amylopectin-synthesizing multienzyme complexes and release PPDK to allow an abundance of PPi in amyloplasts. The high concentration of PPi would then promote plastid AGPS2a and AGPS2b activity and direct the reaction toward ADP-Glc degradation, producing more Glc-1-P for lipid biosynthesis. At the same time, PPDK would actively increase transcription of the *SSIIb* and *SSIIc* genes without increasing starch content. This supports the view that the catalytic activity of one or more of the starch synthase isoforms is insufficient to accommodate the excess ADP-glc present in the amyloplast, but that other constraints within the amyloplast stroma control carbon flux into starch [55]. Further research is required to test this hypothesis. 

The number of spikelets per panicle substantially increased (by approximately 32.5%) in the Suweon 542 mutant (*flo4-4*) compared with the wild type [33]. In the flo4-5 mutant, the number of spikelets per panicle (Table 1) increased only slightly. This may have been due to environmental effects on yield, whereby the average high temperature during the first 4 days after pollination must reach ~33 °C for yield increase to occur [56]. AGPase, the rate-limiting starch biosynthetic enzyme, is a heterotetramer composed of two identical small and two identical large subunits. The enzyme is allosterically controlled, and the isoform in the cereal endosperm is heat-labile [57]. In maize, the *Shrunken*-2 (*Sh*2) gene encodes the large subunit of endosperm AGPase, and overexpression of the gene enhances heat stability and reduces phosphate inhibition, leading to an increase in maize yield (up to 64%) via an increase in seed number [56]. The small subunit of the maize endosperm AGPase is encoded by the gene *Brittle*-2 (*Bt*2), and overexpression of this enzyme also increases yield by about 35%, albeit by an increase in seed number rather than seed weight. This increase is dependent on the average daily high temperature reaching ~33 °C during the first 4 days after pollination [57]. Furthermore, expression of an altered large subunit of endosperm AGPase increases potato tuber yield by 35% [58], wheat yield by 38% [59], and rice yield by 23% [60]. Notably, the yield increases in maize, rice, and wheat are due to increases in seed number rather than increases in individual seed weight. In rice, increases in seed number are a consequence of an enhanced probability of seed development rather than an increase in the number of ovaries. In wild-type rice, only about half of the ovaries develop into fully mature kernels. The remaining kernels cease development, and their contents disintegrate [56,57]. Overexpression of the small subunit of the maize endosperm AGPase (ZmBt1) in rice enhances ADP-Glc synthesis and imports into amyloplasts, but does not lead to further enhancement in seed weight, even under elevated CO_2_ [55]. These observations suggest that the PPDK mutant might enhance the expression of *AGPS2a* and *AGPS2b*, as with AGPase transgene expression, leading to increases in the expression of *AGPS2a*, *AGPS2b*, *SSIIb*, and *SSIIc,* and an increase in seed number rather than seed weight. Our results suggest that enhanced synthesis of ADP-Glc in the rice seed endosperm may increase seed number. Future research regarding the metabolic mechanisms underlying the increase in seed number will be useful for breeding programs aimed at improving rice yield.

## 5. Conclusions

Our data revealed that a novel SNP (C to T) in the coding region (exon 2) of the gene encoding cytosolic PPDK (cy*OsPPDK*) was responsible for the floury endosperm characteristics of low grain hardness, low starch damage, and fine particle size, and this mutation may be valuable in rice breeding programs. Co-segregation analysis with the developed CAPS marker (flo4-5_F/flo4-5_R) revealed co-segregation between the floury phenotype and the *flo4-5* using the segregation population and 44 japonica varieties. This CAPS marker could be applied directly to MAS. Real-time RT-PCR experiments revealed that PPDK was expressed at considerably higher levels in the *flo4-5* mutant than in the wild type during the grain filling stage. Plastids *AGPS2a*, *AGPS2b*, *SSIIb*, and *SSIIc* also exhibited enhanced expression in the *flo4-5* mutant. Although more studies are needed to fully understand the functions of PPDK, our data indicate that PPDK is involved in the endosperm development function through directly or indirectly regulating *AGPS2a*, *AGPS2b*, *SSIIb*, and *SSIIc*.

## Figures and Tables

**Figure 1 genes-11-00465-f001:**
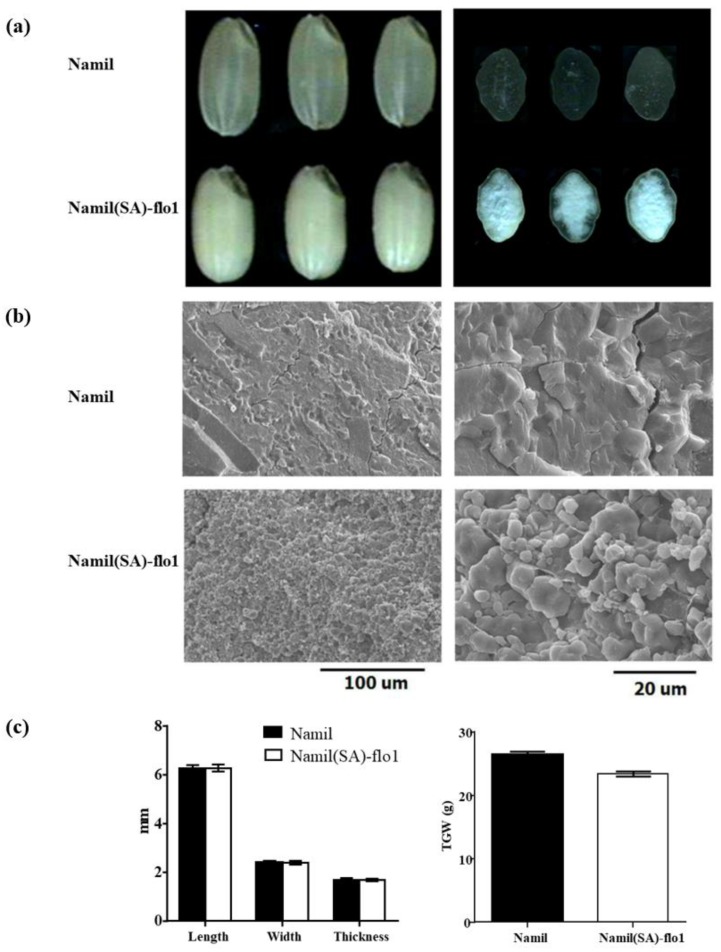
Phenotypic analyses of the mutant Namil(SA)-flo1 and wide type Namil. (**a**) Brown rice and transverse sections of Namil(SA)-flo1 and Namil; (**b**) Electron microscope visualization of mature endosperm. The Namil(SA)-flo1 is packed loosely with compound starch granules; (**c**) The grains’ shape and weight (TGW: 1000-grain weight of brown rice).

**Figure 2 genes-11-00465-f002:**
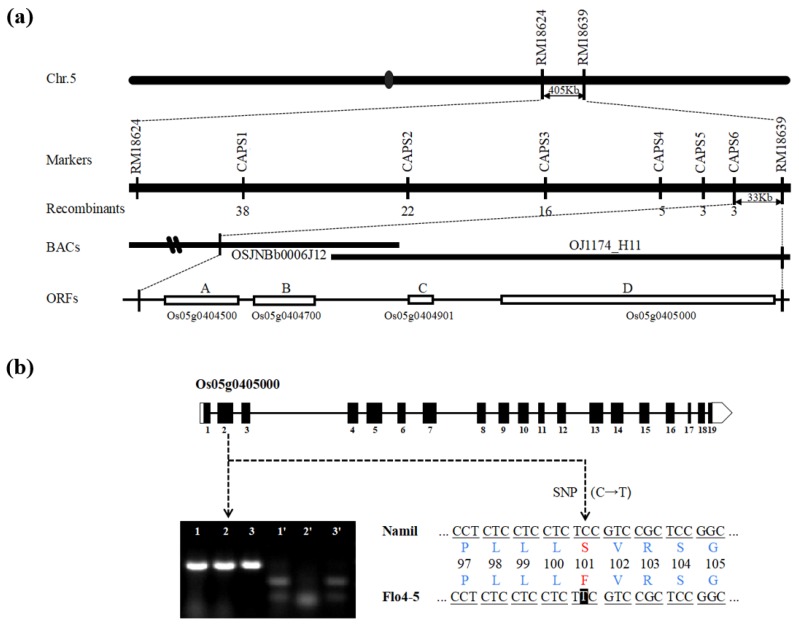
Map-based cloning of the *flo4-5* mutant. (**a**) Fine mapping of the *flo4-5* locus. The molecular markers and number of recombinants are shown. The 33kb virtual contig, composed of overlapping 2 BAC clones, was delimited by e-Landings of two significant markers on the reference rice genome, ‘Os-Nipponbare-Reference-IRGSP-1.0’; (**b**) *flo4-5* gene structure and cDNA sequence comparison showing a nucleotide mutant (C to T) within exon 2 where Ser-101 of the wild was induced to Phe-101 of the *flo4-5*. White boxes represent untranslated regions, black boxes represent coding regions, and solid lines represent introns. 1,2,3 are the PCR results of wild type Namil, *flo4-5*(Namil (SA)–flo1) and *flo4-4*(Namil (SA)–flo2), respectively. 1’, 2’, 3’ are the digested results.

**Figure 3 genes-11-00465-f003:**
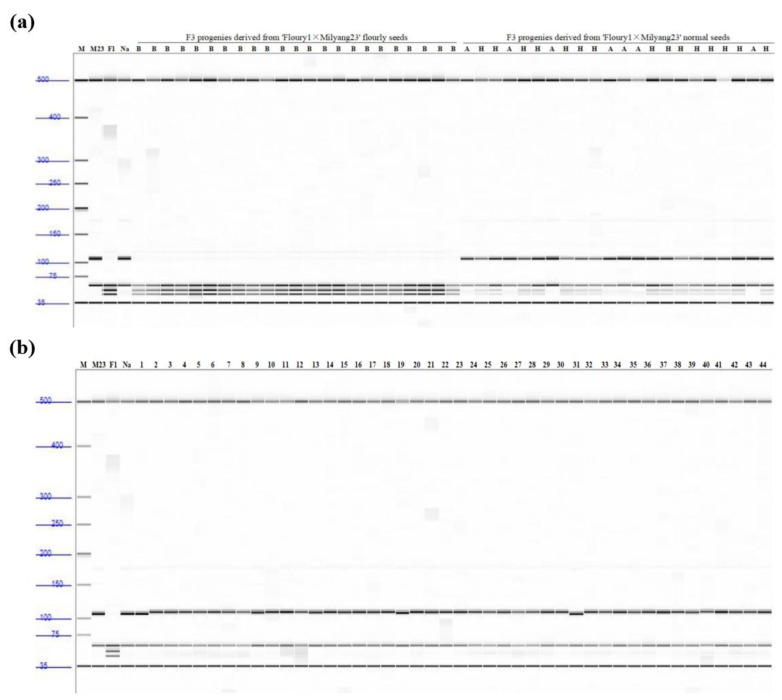
Co-segregation analysis of the *flo4-5* genotype with floury endosperm phenotype. (**a**) Verification of the CAPS marker (digestion by *MboII*) and tagging the *flo4-5* locus using a part of F3 individuals; (**b**) Verification of the CAPS marker (digestion by *MboII*) and tagging the *flo4-5* locus using Korean rice cultivars. ‘A’ and ‘B’ are homogeneous of Namil(SA)-flo1 and Milyang 23, respectively. ‘H’ is heterozygote. M (100bp ladder), F1 (Namil(SA)-flo1), M23 (Milyang 23), Na (Namil). 1. Namil(SA)-flo2(Suweon542), 2. Aranghyangchal, 3. Baegjinju1, 4. Baekogchal, 5. Boramchal, 6. Boramchan, 7. Borami, 8. Boseog, 9. Boseogchal, 10. Boseogheugchal, 11. Cheongnam, 12. Chindeul, 13. Chucheong, 14. Dabo, 15. Danmi, 16. Danpyeong, 17. Deuraechan, 18. Dodamssal, 19. Dongjin, 20. Dongjin1, 21. Dongjinchal, 22. Geonganghongmi, 23. Geonyang 2, 24. Goami, 25. Goami 2, 26. Goami 4, 27. Haepum, 28. Haiami, 29. Hanam, 30. Hanmaeum, 31. Heugjinmi, 32. Heughyangchal, 33. Heugjinju, 34. Heugnam, 35. Heugseol, 36. Hongjinju, 37. Hopum, 38. Hwanggeumnuri, 39. Hwaseong, 40. Hwawang, 41. Hwayeong, 42. Hyangnam, 43. Hyeonpum, 44. Ilmi.

**Figure 4 genes-11-00465-f004:**
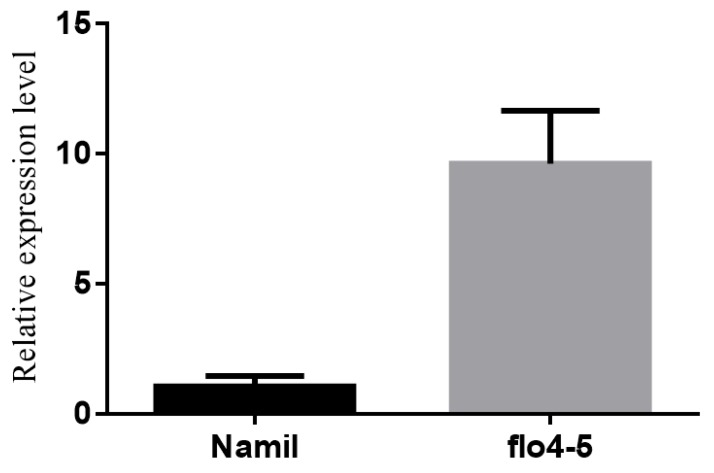
Expression analysis of PPDK between Namil and *flo4-5* at 12 DAF. Values shown are mean ± SD (n = 3).

**Figure 5 genes-11-00465-f005:**
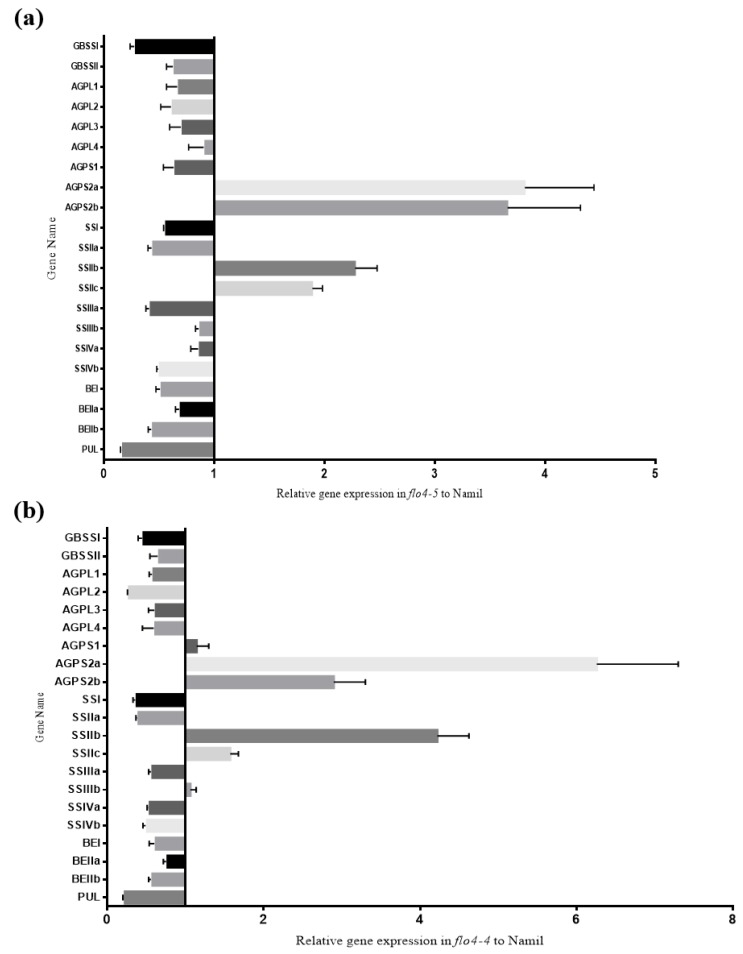
Expression profiles of the genes involved in production of storage starch in endosperm from the flo4-5 mutant (**a**) and Suweon542 (**b**) with the wild type Namil (bar begin with 1). Reference gene was Actin, and the expression levels of these genes are shown relative to the wild type Namil, which is set as 1. Each gene name is indicated by a simplified representation. *GBSSI* and *GBSSII*, granule bound starch synthase I and II; AGPL (*AGPL1*, *AGPL2*, *AGPL3* and *AGPL4*) and AGPS (*AGPS1*, *AGPS2a* and *AGPS2b*), ADP-glucose pyrophosphorylase large subunit and small subunite, respectively; *SSIIa* and *SSIIIa*, soluble starch synthase *IIa* and *IIIa*, respectively; *BEI* and *BEIIb*, branching enzyme *I* and *IIb*, respectively; *PUL*, pullulanase.

**Table 1 genes-11-00465-t001:** Major agronomic traits of Namil(SA)-flo1 in comparison with its wild type, Namil.

Line	HD (Days)	CL (cm)	PL (cm)	TN (No.)	SN (No.)	RGP (%)
Namil	101 ^b^	78 ^b^	25 ^a^	11 ^a^	117 ^a^	87 ^a^
Namil(SA)-flo1	104 ^a^	82 ^a^	25 ^a^	10 ^a^	120 ^a^	74 ^b^

^a^ Means with the same letter are not significantly different at *p* < 0.05 in the least significant difference test (LSD0.05). ^b^ The means of each line were obtained from replicated yield trials with three replication plots. HD: days-to-heading after sowing, CL: culm length, PL: panicle length, TN: tiller number, SN: spikelet number per panicle, RGP: ripened grains percentage.

**Table 2 genes-11-00465-t002:** Physicochemical properties of grains and rice flours.

Line	Hardness Index	Grain Hardness (Kg)	Mean Particle Size (µm)	Damaged Starch (%)	Lightness (CIE Value)	Ash (%)	Protein (%)	Amylose (%)	Lipid (%)
Hwaseong	1.04 a	7825 a	112.2 ± 0.40 a	10.3 ± 0.19 a	88.6 ± 0.01 b	0.84 ± 0.02 a	7.5 ± 0.16 c	18.5 ± 0.24	-
Seolgaeng	0.79 b	5962 b	97.6 ± 1.63 c	7.1 ± 0.10 c	90.3 ± 0.06 a	0.72 ± 0.01 c	6.6 ± 0.11 d	17.5 ± 0.60	-
Namil	1.00 a	7526 a	109.1 ± 0.62 b	9.2 ± 0.17 b	88.7 ± 0.12 b	0.82 ± 0.01 a	9.2 ± 0.25 a	17.7 ± 1.34	1.5 ± 0.56 b
Namil(SA)-flo1	0.45 c	3417 c	86.1 ± 0.81 d	5.1 ± 0.06 d	90.4 ± 0.09 a	0.77 ± 0.02 b	7.8 ± 0.04 b	17.8 ± 0.27	3.1 ± 0.54 a

Note: Different letter means indicate significant differences according to the Duncan multiple range test (*p* < 0.05).

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
