# Peer review of "A New SNP in Rice Gene Encoding Pyruvate Phosphate Dikinase (PPDK) Associated with Floury Endosperm"

_genes, 2020, doi:10.3390/genes11040465_

Round 1

Reviewer 1 Report

Dear Editor/Authors,

Rice is valued worldwide for its nutrient ingredients but it is the most popular in Asia. In this manuscript entitled “A new SNP in rice genes encoding pyruvate phosphate dikinase (PPDK) associated with floury endosperm” authors identified a rice PPDK mutation conditioning the starch-protein balance. PPDK also plays an important role in grain development. The authors presented the impact of PPDK on modulation of endosperm metabolism and this is important information primarily for breeders, due to the high demand for gluten-free products.
The discussion of the results is conducted transparently and concisely.

1) The authors showed that the new SNP in PPDK C is T SNP in exon 2 of cyOsPPDK is responsible for the grain floury characteristics. To date, simple markers based on single nucleotide polymorphism (SNP) discovery to evaluate quality of rice are still limited and little is known regarding the molecular properties of rice PPDK.
2) The authors suggest that enhanced synthesis of ADP-Glc in the rice seed endosperm may increase seed number what can used in breeding programs.
3) The introduction, results and discussion section is clear and concisely.
In Plant material the authors should explain in more details the culture conditions.

Author Response

Review 1

Rice is valued worldwide for its nutrient ingredients but it is the most popular in Asia. In this manuscript entitled “A new SNP in rice genes encoding pyruvate phosphate dikinase (PPDK) associated with floury endosperm” authors identified a rice PPDK mutation conditioning the starch-protein balance. PPDK also plays an important role in grain development. The authors presented the impact of PPDK on modulation of endosperm metabolism and this is important information primarily for breeders, due to the high demand for gluten-free products.
The discussion of the results is conducted transparently and concisely.

1) The authors showed that the new SNP in PPDK C is T SNP in exon 2 of cyOsPPDK is responsible for the grain floury characteristics. To date, simple markers based on single nucleotide polymorphism (SNP) discovery to evaluate quality of rice are still limited and little is known regarding the molecular properties of rice PPDK.
2) The authors suggest that enhanced synthesis of ADP-Glc in the rice seed endosperm may increase seed number what can used in breeding programs.
3) The introduction, results and discussion section is clear and concisely.
In Plant material the authors should explain in more details the culture conditions.

Reply: Thanks for the suggestions. We have made the description more detail about the culture conditions at plant material part.

(Please see Page3, line 139 ~ 152 for details)

Reviewer 2 Report

The study proposed by Wang et al. explores the floury endosperm type in rice (a type of seed appreciated in the industry) and describes the discovery of a mutation in a gene involved in this specific trait.

The authors provide a detailed analysis of the phenotype, of the gene involved and of associated proteins that could be responsible for the floury endosperm type of of Namil(SA)-flo1 with:

  1. Phenotypic description with pictures and SEM showing the distinct internal phenotype of Namil(SA)-flo1 as compared to wild Namil
  2. Fine-mapping candidate genes in a 33kb region, identification of the mutation in PPDK gene as best candidate
  3. Cosegragetation validating the link between the mutation and the phenotype
  4. Functional analysis with qRT-PCR showing the proteins involved in that phenotype and the similarities with a close mutant, Namil(SA)-flo2
  5. Demonstrating the novelty of this mutation with a comparison with modern cultivars

The results appear to be solid and the applications are promising. The methods seem to be technically sound and the conclusions are clear. Thus this study provides an advance in the understanding and control of the floury phenotype and deserves to be published.

To me, the manuscript can be further improved with some restructuring, by providing some more details on the methods and by better discussing some aspects of the results.

Major comments:

  • The structure of the abstract could be strongly improved. It starts right away with conclusions and there is no explanation of the goal of the study and research hypotheses.
  • The introduction is lengthy and this is in stark contrast to the discussion. Maybe some mechanistic details should be moved in the discussion as elements for addressing the specific pathways in the light of the results. Also a long part of the introduction is descriptive and only a short part explains the goal and relevance of the study
  • In the methods population/sample sizes are not always clear. For instance how many plants/plots were measured for assessing the agronomic traits?
  • It is unclear whether there is one or multiple mutations involved in the phenotype. Do you, or can you, state that there is only one candidate gene? The plural form of "gene" in the title reinforces this interrogation.
  • There should be supplementary data provided on result part 3.3 showing how the fine-mapping led to the 33kb region. I would have expected a table with the genotypes obtained with the different markers and the corresponding phenotypes.
  • Overall, the English writing is clear and understandable and I do not have major difficulties understanding the study. However many sentences cannot be published in the present form. There are missing words and incorrect sentences, especially in the discussion. I address this more in details below. Maybe the authors should consider a review by a native English speaker.
  • One part of the discussion is really not clear Line 395-402. Also, because of the form and content of this paragraph, I do not think that the authors provide a good explanation for the increased expression of PPDK in the mutant.

I provide here more details on major comment and minor comments altogether. I hope the authors can address those without difficulties.

  • Title of the manuscript. Why did you choose to write genes in plural when you describe a single mutation in one gene? If there are other potential mutations (as I think is shortly said in results) it should be stated clearly in the discussion.
  • Abstract: please do not use an abbreviation for MAS at this stage. Also,indicate the name of the mutant somewhere.
  • Explain the principle of co-segregation and why you choose this type of experiment
  • This comment refers to previous ones: one interrogation I still have at the end of the paper is whether this unique SNP alone is causing the phenotype. In the results Line 306-309 the authors provide further sequence differences with the mutant and its wild type and these do not appear in the discussion. Or at least it was not clear to me. Can the authors address this gap in the discussion?
  • The distinction between NamilSAflo1 and NamileSAflo2 could be better emphasized in the discussion
  • Use of the terms "glutinuous" and "waxy": please specify at the beginning whether these are exactly the same because you use both "non-waxy" and "non-glutinuous" terms which can be confusing. If the terms are equivalent, consider using only one.
  • If the authors feel they can, it could be very useful to have a graphic representation of the effect of the mutation in the entire process of starch production.
  • You made a functional analysis and this term doesn't appear to describe your work
  • Part 2.2: what are the statistical individuals and the sample size for this range of analyses ? Mention the sample size in each analysis. We need to be able to see how representative are the samples used.
  • Line 169-171: Where the cultivars Hwaseong and Seolgaeng cultivated in the same way and place as the Namil cultivars? This is not really clear from methods.
  • Line 178: the abbreviation for replicated yield trials (RYTs) is not used anywhere, not needed. Also but explain shortly what these replicated trials are (size, number of environments etc..)
  • Line 183: What are the biological replicates here ?
  • I would say that part 2.3 can be merged with 2.2.
  • Line 203: No details are given for the whole genome resequencing data. Please cite appropriately a reference or provide here details about this dataset.
  • Line 219. Please add more information on CLC Viewer (for instance citation or website).
  • Lines 254-255 : "No differences in grain width, length, and thickness were observed between the mutant and the wild type" : please provide result of a test for this claim
  • Line 253:259: the results are given without a clear indication of figure 1 and its different parts, please add.
  • Line 264-265:  “This was consistent with the results of previous reports showing that increasing lipid content could cause floury endosperm. “ this should be moved to the discussion and a reference should be added here.
  • Table 1: indicate that the test is between the 2 groups Namil and Namil mutant
  • Legend Figure 2 please define “E-landings” somewhere, in the methods for instance.
  • Table 1 and 2: Statistical tests LSD and Duncan: please provide the ANOVA result for these posthoc tests. Also add standard deviations for all traits and refer to it in the legend.
  • Line 245: Title: I would replace “phenotypic” by “seed” it is more informative. More generally I would separate pure agronomic traits from grain traits because agronomic traits are not the target traits, while the seed is the main target. This would be more logical.
  • Figure 1c is redundant with the table. You could either delete TGW from the table. On the figure 1c, indicate the letters corresponding to the test to make the result easier to interpret.
  • Line 276-279 refer please to table for these results
  • Line 288-289: Please give few more details about the resequencing and its availability.
  • Line 290-291: How was this multiple comparison done and where are these results? A supplementary table is needed.
  • Figure 3: Refer to lane +1 in the text because there is the ladder in Lane 1.
  • Line 326-327: Add a and/or b to Figure indication.
  • Line 342-343 “was” is lacking in the sentence, I guess.
  • Lines 369-370: “Importantly, the flo4-5 mutant promoted the activity of AGPS2a, AGPS2b, SSIIb, and SSIIc, which might be predicted to lead to high SNs and floury endosperm in the mutant “ This sentence belongs to the discussion and needs a reference.
  • 391:392 : “However, the physicochemical characteristics of flo4-5 is different from the reported flo4-1, flo4-2 and flo4-3” How different are they? I do not see it described here or in the results.
  • 417-418: I do not understand this sentence
  • Line 434 “release PPDK to generate an abundance of PPi in amyloplasts.” This sentence implies a purpose, which is bioligically wrong. Prefer “allowing to” for instance.
  • 442: please write the number of spikelet fully as you do for the next sentence, this will help the reader.
  • 442-469: this paragraph appears to be disproportionally long with regards to the main results and key message of the paper. Please try to simplify this and emphasize other aspects of your study such as the novelty of your results.

Author Response

We revised the manuscript according to the reviewer's comments. Also, we provide a point-by-point response. 

Reviewer 3 Report

Major Comments.

In general, I find the aim of the study is not well defined and the text is quiet confusing regarding the purpose of investigation.

Line 98 the authors write- Seven 
 floury endosperm genes (FLO1–7) have been identified in rice to date.

The text gives an impression that they have two major aims:

  1. Biochemical characterization of flo1 mutant in rice.

2 fine mapping of an allelic variant of flo4 (FLO4-5).

However, the link between the two is not justified.

 It would help to clearly define the aims of the study and clearly write whether referring to a gene or to an allele of mutant for a given gene.

For example- in line 154-“ In this study, the agronomic traits and grain physicochemical properties, including suitability 
of Namil(SA)-flo1 were investigated.

And the next line hints towards identifying the causal gene in flo-4 but that’s an allele presumably for OsPPDK (cytosolic version).

Second important aspect missing in the paper is the similar biochemical analysis for flo4-5 allele that they have characterized the underlying gentic cause (C-T transition).

Minor Comments.

  1. Gene names to be italicized throughout the text.
  2. Line 33- “……mutation may be valuable in extending of rice utility” perhaps better to rephrase.

  3. Line 101- Please spell out TPR and CBM. This applies to many abbreviations used throughout the text.
  4. Line 124- https://link.springer.com/article/10.1023/A:1005884515840. This seems an important citation that authors may have missed.
  5. Line 311 fix the reference. (Wang et al 2018)
  6. The legends need more description as it’s not clear what the images point to.

Author Response

Major Comments.

  • In general, I find the aim of the study is not well defined and the text is quiet confusing regarding the purpose of investigation. Line 98 the authors write- Seven floury endosperm genes (FLO1–7) have been identified in rice to date. The text gives an impression that they have two major aims:

Biochemical characterization of flo1 mutant in rice.

fine mapping of an allelic variant of flo4 (FLO4-5).

However, the link between the two is not justified.

  • It would help to clearly define the aims of the study and clearly write whether referring to a gene or to an allele of mutant for a given gene. For example- in line 154-“ In this study, the agronomic traits and grain physicochemical properties, including suitability 
of Namil(SA)-flo1 were investigated. And the next line hints towards identifying the causal gene in flo-4 but that’s an allele presumably for OsPPDK (cytosolic version).

Response:  we revised that part to make it simple and to be clear

(Please see line 129-130 for details.)

  • Second important aspect missing in the paper is the similar biochemical analysis for flo4-5 allele that they have characterized the underlying gentic cause (C-T transition).

Response: Thanks for the comments. In this study, we identified a novel SNP (C to T) in the coding region of FLO7 (PPDK), generated an entirely milky-white opaque kernel except for a thin peripheral area of the grain and the maturity starch granules are packed loosely with irregular and rounded shape, was derived from sodium azide mutagenesis of Namil. The mutant genotype and phenotype of floury endosperm are inconsistent with the reported of a T-DNA insertional knockout mutant of the rice PPDK gene (flo4) produce an opaque seed. Furthermore, FLO7 with a continuous expression during the grain filling stage was responsible for the development of an entirely milky-white opaque kernel, it different from the flo4 showed that rice with inactivated PPDK gene failed to produce an opaque seed. Taken together, this discovery suggested that a new floury endosperm mutant makes the grains attractive for use in the food industry and provides further insight into the floury endosperm of rice.

Minor Comments.

  • Gene names to be italicized throughout the text.

Response: Thanks for the correction. We have updated these in the whole text.

  • Line 33- “……mutation may be valuable in extending of rice utility” perhaps better to rephrase.

Response: we deleted it.   

  • Line 101- Please spell out TPR and CBM. This applies to many abbreviations used throughout the text.

Response: Thanks for the comments. We have reedited the abbreviations in the whole text.

(Please see Page 2, line 75 and 76 for details)

  • Line 124- https://link.springer.com/article/10.1023/A:1005884515840. This seems an important citation that authors may have missed.

Response: Thanks for the suggestion. We have cited it as a reference.

(Please see Page 2, line 98)

  • Line 311 fix the reference. (Wang et al 2018)

Response: Thanks for the corrections. We have re-edited it and revised the reference part carefully.

  • The legends need more description as it’s not clear what the images point to.

Response: Thanks for the suggestion. We have made the description more detail at all legends.

Reviewer 4 Report

The description of sequencing and sequence analysis is incomplete and unclear. Methods does not mention Illumina or any other sequencing technology, whether any read trimming or sequence quality control was used, what sequence alignment software was used, or whether any SNP calling software was employed. If the SNP was inferred visually, then a screenshot of the sequence alignment should be included in supplement. The report, “Multiple comparisons were conducted between the genotypes of these recombinants and the phenotypes of their offspring” (line 290) is vague and meaningless. The statement, “the whole genomes of Namil(SA)-flo1 and Milyang 23 were re-sequenced and aligned against one another” (line 287) is misleading because there was no whole-genome re-sequencing; instead genomic sequence was cloned, and the clones were sequenced, most likely. Methods says, “Complete genomic DNA of the cyOsPPDK gene was cloned in three overlapping segments using primers designed on the basis of the cyOsPPDK gene sequence of Nipponbare” (line 229). Again, the phrase “complete genomic DNA” muddies the story; most likely, one gene was cloned from two genomes. The paper does not mention if the raw reads were submitted to a public database. The GenBank accessions should be hyperlinked. In case it helps, here is my most optimistic interpretation of the sequencing and sequence analysis section (lines 284-313). One candidate region was cloned from the mut and wt genomes; the two clones were sequenced and the sequence reads were mapped to reference BAC sequences; four candidate genes were identified along the BAC using a genome viewer; all four candidates were sequenced again from the mut and wt clones; reads were mapped to the Nipponbare reference; one SNP was identified; for the gene that contained the SNP, cDNA sequences were inferred from mapped reads; the two inferred cDNA sequences were submitted to GenBank. Here is perhaps a comparable paper that was published in Genes with sequence alignments and more provide in the supplement: Development of Molecular Marker Linked with Bacterial Fruit Blotch Resistance in Melon (2020) Genes doi:10.3390/genes11020220

The Abstract presents a more certain picture than does the Discussion. According to Discussion, the authors are putting forward a hypothesis: “Based on previous research and our findings, we hypothesize that a single amino acid change (S to F) in the PPDK protein in the flo4-5 mutant may disrupt the amylopectin-synthesizing multienzyme complexes and release PPDK to generate an abundance of PPi in amyloplasts” (line 432). But according to the Abstract and the Conclusions, the data revealed this to be a fact: “Our data revealed that a novel SNP (C to T) in the coding region (exon 2) of the gene encoding cytosolic PPDK (cyOsPPDK) was responsible for floury endosperm characteristics and this mutation may be valuable in extending of rice utility.” These two versions should be reconciled.

The Abstract and the Conclusions say, “our data indicate that PPDK modulates endosperm metabolism”. That phrase appears nowhere else in the paper so it is hard to find the specific evidence for that finding.

The Abstract and the Conclusions say, “PPDK modulates endosperm metabolism through regulation of AGPS2a, AGPS2b, SSIIb, and SSIIc.” The Introduction does not provide background on these. The manuscript does not provide even a hypothesis for why these were tested.

Running over two pages with 44 references, the Introduction is slow to get to the point. Apparently, it does not cover the necessary basics, because the authors provide additional background in Methods and Results (e.g. “Rice flour is a popular ingredient,” line 274). Is this vast review necessary for anyone who is going to read this paper? In case it helps, here is my summary in two sentences. The molecular mechanisms regulating floury endosperm in rice remain unknown. The previously discovered FLO4-5 mutant remains uncharacterized and the underlying mutation was previously unknown.

The term CAPS, used extensively, is barely explained and not referenced (e.g. line 203).

The PROVEAN tool is mentioned twice without a proper reference (e.g. line 233).

The caption to Figure 1c should specify sample size N.

The verb hypothesize should be the noun hypothesis in line 399.

Author Response

We revised the manuscript according to the reviewer's comments. Also, we respond to point-by-point. 

Reviewer 5 Report

In general, the paper is well represented. There are several minor changes that can be addressed so the paper can be improved.

  1. On Results section, 3.3. Fine mapping of the floury endosperm locus; line 299: "In rice, PPDK encodes both a cytosolic and a chloroplast-targeted PPDK protein (cyOsPPDK and chOsPPDK, respectively)". 
    This sentence is a bit confusing for ones who are not familiar with rice PPDK. Perhaps you can mention gene loci encoding cytosolic and chloroplast-targeted PPDK proteins. Or are they encoded by Os05g0405000?

  2. On Results section, 3.4.Co-segregation and expression analyses; line 337-338 about the PROVEAN tool. What does it mean by 'deleterious'? I think the SNP does not cause any stop codon, but may cause a change in protein structure. Checking whether the protein structure changes may help explain the association of SNP in exon 2.
    Also, is the increase of PPDK transcript level expected in flo4-5 mutant?

  3.   On Discussion section, page 395-396: "It is unexpected that transcript expression in the flo4-5 mutant". The sentence is truncated.

Author Response

Review 5

In general, the paper is well represented. There are several minor changes that can be addressed so the paper can be improved.

  • On Results section, 3.3. Fine mapping of the floury endosperm locus; line 299: "In rice, PPDK encodes both a cytosolic and a chloroplast-targeted PPDK protein (cyOsPPDK and chOsPPDK, respectively)". 
    This sentence is a bit confusing for ones who are not familiar with rice PPDK. Perhaps you can mention gene loci encoding cytosolic and chloroplast-targeted PPDK proteins. Or are they encoded by Os05g0405000?

Response: Thanks for the corrections. We have made the description more detail in line 273-279.

  • On Results section, 3.4.Co-segregation and expression analyses; line 337-338 about the PROVEAN tool. What does it mean by 'deleterious'? I think the SNP does not cause any stop codon, but may cause a change in protein structure. Checking whether the protein structure changes may help explain the association of SNP in exon 2.
    Also, is the increase of PPDK transcript level expected in flo4-5 mutant?

Response: Thanks for the comments. In here, “deleterious” means that the single nucleotide mutation (C to T) on the FLO4-5 gene might change the structure of the protein and affect its activity. In results, our data indicate that the mutation site could active the functional of PPKD. The result is consistent with the reported that releasing PPDK would increase lipid synthesis and altered physicochemical properties (Zhou et al. 2016).

  • On Discussion section, page 395-396: "It is unexpected that transcript expression in the flo4-5 mutant". The sentence is truncated.

Response: Thanks for the comments. We corrected it.

Round 2

Reviewer 3 Report

Figure 1 and Table 2 summarises physiochenical properties of flo1 mutant while lines 353-357 describe them as flo4-5.

Line 351- Please italicise flo4-5

Manuscript will benefit from improving the written english.

Author Response

Figure 1 and Table 2 summarises physiochenical properties of flo1 mutant while lines 353-357 describe them as flo4-5.

  • Response: Thanks for the corrections. We have reedited it.
    (Please see Page 12, line 374-389 for details)

Line 351- Please italicise flo4-5

  • Response: Thanks for the corrections. We have reedited it.
    (Please see Page 12, line 374-389 for details)

Manuscript will benefit from improving the written english.

  • Response: Thanks for the suggestions.  We have revised the current manuscript according to MDPI English Editing service. We attach the certificate. 
